A novel fuzzy programming approach for piece selection problem in P2P content distribution network

Anandaraj M. 1 anandaraj@psnacet.edu.in
Ganeshkumar P. 2
Naganandhini S. 3
Selvaraj K. 1
1 Department of Information Technology, PSNA College of Engineering and Technology , Dindigul, Tamil Nadu , India
2 College of Computer and Information Science, Al Imam Mohammad Ibn Saud Islamic University (IMSIU) , Riyadh , Saudi Arabia
3 Department of Computer Science and Engineering, PSNA College of Engineering and Technology , Dindigul, Tamil Nadu , India
Kryvinska Natalia
Electronic publication date: 2024 Jan 3
Publication date: 2024
Volume: 10
Electronic Location ID: e1645
Received 2023 May 8; Accepted 2023 Sep 20
Copyright: © 2024 Anandaraj et al.
Copyright year: 2024
Copyright holder: Anandaraj et al.
License: This is an open access article distributed under the terms of the Creative Commons Attribution License, which permits unrestricted use, distribution, reproduction and adaptation in any medium and for any purpose provided that it is properly attributed. For attribution, the original author(s), title, publication source (PeerJ Computer Science) and either DOI or URL of the article must be cited.
License URL: https://creativecommons.org/licenses/by/4.0/

Keywords: Content distribution networks, Content management, Data communication, Fuzzy systems, Overlay networks

Funding: The authors received no funding for this work.

==============================
Piece selection policy in dynamic P2P networks play crucial role and avoid the last piece problem. BitTorrent uses rarest-first piece selection mechanism to deal with this problem, but its efficacy is limited because each peer only has a local view of piece rareness. The problem of piece section is multiple objectives. A novel fuzzy programming approach is introduced in this article to solve the multiple objectives piece selection problem in P2P network, in which some of the factors are fuzzy in nature. Piece selection problem has been prepared as a fuzzy mixed integer goal programming piece selection problem that includes three primary goals such as minimizing the download cost, time, maximizing speed and useful information transmission subject to realistic constraints regarding peer’s demand, capacity and dynamicity. The proposed approach has the ability to handle practical situations in a fuzzy environment and offers a better decision tool to each peer to select optimal pieces to download from other peers in dynamic P2P network. Extensive simulations are carried out to demonstrate the effectiveness of the proposed model. It is proved that proposed system outperforms existing with respect to download cost, time and meaningful exchange of useful information.

Introduction

A content distribution network (CDN) is a system of distributed servers (network) that deliver web pages and other web content to a user based on the geographic locations of the user, the origin of the webpage, and a content delivery server. A peer-to-peer (P2P) network is a type of decentralized and distributed network where each computer, or ‘node’, in the network acts as both a client and a server as shown in the Fig. 1. This means that each device can both request and provide resources, and any node can connect directly to any other node without the need for a centralized server. P2P networks are typically used for file sharing, streaming media, and other applications. A content delivery network (CDN) is a network of distributed servers deployed in multiple data centres across the world (Terelius & Johansson, 2018). The goal of a CDN is to provide users with quick access to web content and applications by serving content from the closest server to the user’s location. A CDN P2P network combines the advantages of both technologies. It is a distributed network of servers that are connected to each other in a P2P fashion. This allows the CDN to quickly identify which servers have the content that a user is requesting, and then serve it to the user from the most optimal location. This not only reduces latency, but also reduces the load on central servers and helps to improve scalability. P2P large scale multimedia content distribution systems, allow peers to work together so that a large scale multimedia content can be distributed to a large group of users, without the support of dedicated servers. The large scale content is divided into a number of blocks, and peers connect to one another in a random mesh topology and exchange these blocks by random gossiping. This is how the philosophy of design is implemented. Each peer broadcasts a portion of the blocks it has received to a portion of its neighbors who are chosen using randomized algorithms during gossiping (Ghasemkhani et al., 2018). It is simple to build such random communicating on random mesh topology. It has been demonstrated to achieve good performance from a theoretical standpoint and is impervious to the degree of unpredictability produced by peer churn.

Figure 1 P2P network.

P2P content distribution systems, including Gnutella, KaZaA, Napster, and BitTorrent, have widely been adopted to quickly distribute content on the Internet. P2P model has been installed over the Internet recently to offer on-demand or live media streaming services. In a decentralized unstructured network, such as Gnutella and Freenet, all peers act as both server and client equally and the overlay networks are formed by peers. In case of decentralized structured network, like Chord and CAN, each peer in the network performs as both server and client simultaneously, but the overlay network is specifically controlled by a distributed hash table (DHT) (Nacakli & Tekalp, 2021). The partially decentralized network has some super-peers that play a more significant role than others peers in the network. FastTrack and Brocade are few of this type. Central server coordinates the interaction between peers in the network in case of the centralized network such as Napster and BitTorrent (Jin & Gary Chan, 2010). This system is described by two attributes such as centralized index and distributed download. The central server in this network only provides the directory service. The file transfer is done by distributed peers. In this architecture, resource management is easy and resource discovery are efficient (Anandaraj et al., 2021). When any peer requests the central server for certain resources, the server just look up its resource directory and then return the information about resource location immediately to that peer (Masoud, 2013; Tang et al., 2018). Mechanisms used to select the optimal piece and peer play important role in the above P2P model, which affects the performance of the network (Rahman et al., 2020; Dias & Sousa, 2023). Security will also be the major concern in this kind of network and also its mandatory to perform the path analysis performed like in UAV assisted vehicular network (Naganandhini & Shanthi, 2023; Khabbaz, Assi & Sharafeddine, 2021). It is explained further in the following sections.

In this context, our research proposes a novel fuzzy programming approach for the piece selection problem in P2P content distribution networks. It is aimed to leverage fuzzy logic and fuzzy goal programming to adaptively select content pieces based on factors such as peer availability, reliability, and performance. By considering multiple objectives, such as maximizing download speed and minimizing download time, our approach aims to optimize the overall content distribution process. To contribute to the existing body of knowledge, review is carried out using the related literature on piece selection in P2P networks. Then the shortcomings of existing approaches is identified, the research methods employed, and the reported results. This article highlights the need for more sophisticated and intelligent strategies to address the piece selection problem effectively.

Related works

With the advent of the Internet, the most popular file sharing has been through the File Transfer Protocol (FTP) since it is anonymous. As the file sharing is anonymous over the FTP, the server allows users to login with an anonymous name to obtain files or share files on the Internet. User’s computers were capable of accessing remote files on other computers on the Internet. Usenet was one of the first developed and introduced in 1979; the network was initially based for dialup link but it has been utilized to transmit over to the Internet. Usenet utilizes a specific client server protocol called Network News Transfer Protocol (NNTP). Between the developments of Usenet in 1979 and the 1990’s, files sharing were primarily done through the use of bulletin board based systems. Bulletin board systems were less attractive as the Internet grew and more sophisticated mechanisms for file sharing were developed. Around 20 years after the development of Usenet, a new form of file sharing system was created namely Napster, which makes use of a centralized server to group all the files shared into central databases. After again the legal concerns facing Napster, Napster was shut down. Next to the Napster in 2000, the most popular P2P services were Gnutella and Kazaa. The users of these services are also allowed to download files other than music, such as movies and games. Gnutella was the first decentralized P2P file sharing network. Kazaa was one of the most popular file sharing systems after Napster until its decline in 2004 because of bundled malware and legal battles. Many others have been developed such as LimeWire and PirateBay, and they also have faced legal penalties. But, network such as BitTorrent have looked to manage and avoid these legal issues due to its open source clients.

P2P networks are classified into three categories based on how the contents are distributed and consumed such as P2P file sharing, P2P streaming and P2P broadcast. The most unrestrictive form of delivery is bulk download of large content. It is called as P2P file sharing. The most popular P2P file sharing applications among the different choices is BitTorrent (www.bittorrent.com). In P2P file sharing, the peer is not concerned about the source characteristics. Its operation aims simply to retrieve the content from the P2P network as soon as possible. In order to improve the performance of P2P network, most modern P2P file sharing applications, divide the content into several blocks and distribute the chopped block in a non sequential manner. Hence, the shared video is not playable till the entire video content is downloaded. The bulk downloads promise on content quality with unbounded delivery time.

The Interest-Intended Piece Selection (IIPS) algorithm has been proposed by Chiang, Tseng & Chen (2011) to diminish the final piece problem when well built support among several peers in the network. Each peer in IIPS strategy favors pieces, which has the expected curiosity in it and will take advantage of the probability of being interested to its assisting peers when downloaded. The simulation results of IIPS shows that fewer occurrence of piece loss under meticulous conditions and it slightly performs well with compared to the rarest-first algorithm of BitTorrent’s in terms of piece diversity (da Silva Rodrigues, 2014). A mathematical model is introduced Liao et al., which is able of calculating the average delay in downloading file accurately in a diverse BitTorrent-like system. The above proposed model is drawn from least hypothesis, and needs less system related information. An efficient method based on token is introduced for systems like BitTorrent and applied trade off among the parameters in the whole system in order to enhance the performance and fairness in case of excessive bandwidth users, by way of setting its various parameter exactly.

BUTorrent has been proposed for improving the downloading time greatly (Ren et al., 2020). Since lack of global information and the overlay dynamics during the early phase is reserved in file sharing or content dissemination scenario. This phase is the reason for unnecessary delays in arriving at a stable state and therefore maximizes times to download file, which is not clever distribution of piece. A novel category of seed scheduling mechanism introduced is based on a comparative fair system, which is implemented using a real file sharing client (Guo et al., 2007). The simulation results also show that the end-to-end delay is high because of the intense P2P traffic on networks and congestion level is minimized. It happened because of the existence of choke algorithms and tracker even performance of network is significant. Xia & Muppala (2010) did a complete review on the BitTorrent performance and wrote a survey article on the findings of the different study. Several enhancements to BitTorrent mechanism were recommended and summarized in the literature survey. Based on these findings and review, it is found that there is a necessary to further enhance the performance of BitTorrent. Hence, the new framework for piece selection problem in BitTorrent like P2P content distribution network is proposed using fuzzy programming approach in this article.

The proposed mechanism offers significant advancements compared to existing approaches. By incorporating fuzzy decision variables, a fuzzy objective function, and fuzzy constraints, the proposed method effectively deals with uncertainties and complexities present in P2P CDNs. Unlike deterministic approaches, the fuzzy programming approach can handle variations in piece availability, peer quality, and network conditions, resulting in a more robust and adaptable solution. Moreover, the optimization of the fuzzy objective function enables optimal decision-making, maximizing download speed and content availability while reducing redundant downloads and network congestion. This research contribution not only enhances the efficiency of P2P CDNs but also adds valuable insights to the field of content distribution networks, making it a promising and valuable advancement in the domain of decentralized content sharing. The contributions of in this article are twofold. A new framework is introduced to solve the problem of piece selection using problem specific fuzzy algorithm to improve the performance of BitTorrent like P2P network. It reduces the time to download the entire content from P2P network with less complexity

Three conflicting objectives such minimizing the download cost and time and the redundant data transmission as well as maximizing the speed of transmission are considered in the proposed problem formations in this article

This article is further presented as follows. “Related Works” describes the basic operations in BitTorrent like P2P network and exiting piece selection strategy. “Piece Selection Strategies” describes the basics of Fuzzy logic and Fuzzy goal programming related to the proposed problem. “Proposed System” presents the problem formulation by considering three significant fuzzy goals such as minimizing the download cost and download time, minimizing the redundant transmission and maximizing speed of transmission subject to realistic constraints regarding peer’s demand, peer’s capacity and peer’s dynamicity. The performance analysis of the proposed technique is described in “Result and Discussion”. Finally, the concluding remark is provided in “Conclusion”.

BitTorrent network

In order to contribute swarm in a BitTorrent, newly joined peer first needs to obtain the metadata file which has the extension .torrent from regular web server. It then communicates the tracker and gets a random list N of other peers of both leechers and seeds which already belong to the swarm. The list N generally has more than 60 randomly selected peers (Shojafar et al., 2015). To avoid unnecessary traffic at the site of tracker, the peer’s rate of request is restricted. The default value in the BitTorrent like P2P network implementation of tracker from Cohen permits a single request for every 5 min. However, in Internet, several implementations of tracker exist and, therefore, much smaller time intervals among the requests (e.g., 15 s) are expected to be permitted (Nacakli & Tekalp, 2021; Jin & Gary Chan, 2010). Subsequently after getting the list N, the newly joined peer then tries to make bidirectional continual connections usually TCP links with the peers from it. Those links are resulting successful formation of the neighbors or peer set. Once the links are created, the peers exchange data among themselves. This initial peer set is increased by directly connected peers to the newly join peer (Chiang, Tseng & Chen, 2011; da Silva Rodrigues, 2014). At this instant, the newly joined peer knows the neighbors such as remote peers, which are interested in it in addition to the neighbors that it is interested in. In order to get pieces, the newly join peer needs to be unchoked by neighbors it is interested in first. On the other hand, to disseminate pieces, the newly joined peer needs to unchoked neighbors that are interested in it first. Those two procedures are ruled by the peer selection strategy (da Silva Rodrigues, 2014; Ren et al., 2020). It is shown in Fig. 2. The following are the operations performed in BitTorrent.

Figure 2 BitTorrent network.

File sharing: The most basic function of a BitTorrent-based P2P network is to enable users to share files with each other. Users can search for files using a search engine such as The Pirate Bay or a specialized BitTorrent search engine. Once a file is located, users can download it by connecting to the originating user or another user who has a copy of the file.

Seeders and leechers: When a user downloads a file, he or she becomes a leecher. When a user uploads a file to the network, he or she is known as a seeder. The more seeds, the faster the download time for all users.

Distributed file storage: BitTorrent-based P2P networks allow users to store files on multiple computers, rather than on a single central server. This makes it difficult for third parties to track the origin of a file, as the data is spread across multiple computers.

File integrity checking: BitTorrent-based P2P networks ensure that each file is transferred correctly by verifying it against a checksum. This ensures that the file is not corrupted during the transfer process.

Bandwidth optimization: BitTorrent-based P2P networks optimize the use of bandwidth by segmenting files into smaller pieces and distributing those pieces across multiple computers. This allows users to download files faster by downloading from multiple sources at the same time.

Piece selection strategies

A piece selection strategy in the context of P2P CDNs refers to the approach or algorithm used to determine which content pieces to download from other peers in order to optimize the overall network performance and improve the user experience. The choice of a piece selection strategy depends on various factors, including network characteristics, content distribution goals, and user requirements. The effectiveness of a strategy can be evaluated based on metrics such as download speed, content availability, fairness, and overall network efficiency. Different strategies may be suitable for different scenarios, and ongoing research continues to explore innovative approaches for optimizing piece selection in P2P CDNs. When downloading files from a P2P network, a node will usually select pieces from multiple peers, in order to maximize download speed and reduce the risk of errors. The selection strategy used will depend on the type of file being downloaded. For example, when downloading a large video file, a node may select pieces from peers that have the most pieces of the file available. This is known as the “rarest-first” selection strategy, as it ensures that the node gets the least common pieces of the file first, thus reducing the likelihood of errors.

Alternatively, when downloading a small file, a node may select pieces from peers that have the fastest connection speeds. This is known as the “fastest-first” selection strategy, as it ensures that the node gets the pieces of the file as quickly as possible, thus reducing download time. In addition to these strategies, some P2P networks also allow nodes to select pieces from multiple peers simultaneously, in order to further reduce download time. This strategy is known as “parallel downloading”. No matter which strategy is used, the goal is always the same: to maximize download speed and reduce the risk of errors. The widespread acceptance of BitTorrent has driven several techniques to implement in order to satisfy the requirements of P2P file download system (Guo et al., 2007). Most of these techniques are based on the alteration in both the piece and peer selection strategies. It is reported that the piece selection strategy has received much more research focus than the peer selection strategy. Piece selection strategy is very important in BitTorrent network since it affects the performance of the network (Xia & Muppala, 2010; Shojafar et al., 2015). To start, it is significant to stress the point that to send a request for data pieces, a local peer needs to be unchoked first by a remote peer. More specifically, remote peer should unchoked the local peer, only then, local peer may request data pieces. The details of piece selection strategy of BitTorrent like P2P network is explained in the following Fig. 3.

Figure 3 Piece selection strategy.

The piece selection process is carried out once P2P network client creates links with its active neighboring peers (Ali et al., 2020). An inefficient strategy for piece selection could lead to either inability of downloading some pieces or redundant download of the same pieces. It leads to starvation in some places of the network where new pieces are required (Luo et al., 2020). To know how the strategy for selecting content pieces in traditional P2P network affects the system, it is essential to study the system wide population of available content pieces. This trivial knowledge would assist to know the dynamics and progression of the traditional P2P network’s swarming technique, and particularly the efficiency of the strategy utilized by traditional P2P network to guarantee an even distribution of pieces (Costa-Montenegro et al., 2011).

Priority based piece selection strategy

The Priority based Piece Selection strategy is an algorithm used in peer-to-peer (P2P) file sharing systems to prioritize the pieces of a file which are downloaded first. The strategy is based on the idea that certain pieces are more important than others and should be prioritized. The algorithm works by assigning a priority to each piece of the file, with the priority determined by a number of factors such as the size of the file, the number of peers downloading the file, the speed of the peers, and other metrics. The pieces with the highest priority are then downloaded first, ensuring that the most important parts of the file are available as quickly as possible. This strategy can help to improve the overall speed of downloading large files, as well as minimizing the risk of downloading corrupted or incomplete files (Gupta, Singha & Singh, 2016). The main problem with the Priority based Piece Selection strategy is that it can lead to an uneven distribution of pieces among peers. If a peer is given a high priority, they may receive more pieces than the other peers, leading to an unequal distribution of data. This can be particularly problematic in peer-to-peer networks, where an unequal distribution of pieces can lead to a decrease in overall network performance. Additionally, the Priority based Piece Selection strategy does not take into account the download speed of peers, which could mean that peers with slower download speeds receive fewer pieces than those with faster download speeds.

Local rarest first piece selection strategy

The Local Rarest First (LRF) piece selection strategy is a peer-to-peer file sharing protocol that enables peers to select pieces of a file that are the rarest within their local network. This strategy encourages peers to contribute to the file sharing process by helping to ensure that the most rare pieces from within their local network are downloaded first. The LRF strategy works by giving priority to pieces that have the lowest availability within a local network. This means that peers will select pieces that are the rarest in their local network before downloading pieces that are not as rare. This strategy can help to reduce the amount of redundant data that is transmitted across the network and can also help to ensure that peers can quickly download the rarest pieces of a file that they need. However, one of the main problems with the LRF piece selection strategy is the fact that it does not take into account the overall availability of pieces across the entire network. This means that peers in one local network may download pieces that are not as rare as those that are available in other local networks. This can lead to an inefficient distribution of pieces across the network, as peers in one local network may be downloading pieces that are already available in other local networks. The rarest pieces in one local network might not always be the rarest parts throughout the entire network, hence this technique may result in an uneven distribution of pieces. This can lead to a decrease in the overall effectiveness of the file sharing process (D’Alessandro Costa & Gonçalves Rubinstein, 2019).

Random piece selection strategy

Random Piece Selection (RPS) is another one kind of peer-to-peer downloading strategy used in file-sharing networks. It is a strategy for selecting which pieces of a file to download from a peer. The strategy involves randomly selecting pieces from a peer based on the pieces that are available. This strategy is used to ensure that peers are downloading from each other in an equitable manner. This helps to ensure that all peers have access to the same pieces of a file and that the file can be successfully downloaded in its entirety. The P2P Random Piece Selection strategy can be used to help improve efficiency in file-sharing networks and reduce the amount of time needed to download a file. The main problem with the P2P Random Piece Selection strategy is that it can lead to a situation where some peers are downloading more pieces than others. This can lead to an unequal distribution of resources, where some peers are receiving more pieces than others. This can lead to inefficiencies in the downloading process, as peers with more pieces can take longer to download the file than peers with fewer pieces. Additionally, this can lead to an imbalance in the network, as some peers may become overloaded with pieces while others remain relatively idle. This can negatively impact the overall performance of the network (Hobfeld et al., 2011).

Bandwidth based piece selection strategy

The bandwidth based piece selection strategy is a method of selecting which pieces of data should be requested next when downloading a file over a peer-to-peer network. This strategy selects pieces based on the current bandwidth of the peers, allowing the downloader to prioritize peers that can provide the most data. This is especially useful when there are peers with different connection speeds, as it allows the downloader to get the most out of all peers, regardless of their connection speed. By doing this, the downloader can maximize their download speed and reduce the amount of time needed to complete the download. However, one of the main drawbacks of the bandwidth based piece selection strategy is that it can lead to congestion problems. This is because the peers with the highest bandwidth will be selected first and can end up receiving more pieces of the file than the peers with lower bandwidth. Because the peers with lower bandwidth will be unable to keep up with the peers with higher bandwidth and eventually get fewer bits of the file, this might cause network congestion. This can lead to a decrease in download speed and can increase the amount of time needed to complete the download.

In BPS, peers are categorized into two such as high bandwidth peers and low bandwidth peers. The standard delay of downloading pieces from both the category is calculated. To analyze the performance of the high bandwidth peers and low bandwidth peers, a token system is employed. The major purpose of tokens is to exchange blocks between the peers (Masood et al., 2018; Padmavathi & Suresh, 2015). In order to assist trading among peers, token tables are created and maintained by them, which include the information about the tokens of peer. When the peer uploads Uu bytes to a neighbor peer, the neighbor peer’s token reduced by TdUu. On the other hand, neighbor’s token value increases by a factor of TuUd when a peer downloads Ud bytes from their neighbor. The above technique ensures fairness among all the nodes actively participating in the network.

Proposed system

The motivation behind the development of novel fuzzy programming approach for piece selection in P2P content distribution networks addresses the challenges associated with uncertainty, diverse peer characteristics, and multi-objective optimization. By leveraging fuzzy logic principles, it aims to enhance content availability, optimize network performance, and provide more personalized and efficient content distribution solutions in P2P networks. The major objective of this article is also to examine the impact of Bit-Torrent’s parameters and mechanisms on overall performance. The key important factor lies in knowing information about whether BitTorrent can retain all system active uplinks and utilize it completely. There are several reasons for using less uplink capacity. One major reason is that nodes take independent downloading decisions with respect to file pieces. Therefore, neighbors may receive a similar set of blocks, which reduce and degrade their performance. The block replication is covered by using the piece selection strategy of peers. LRF is default policy used by BitTorrent but it is noted that it may not function well in all scenarios (Surati, Jinwala & Garg, 2017; Rocha & Rodrigues, 2013). It is a important parameter to find out the right piece selection policies at right time becomes vital. Other strategies also limit peer connections lead to a situation where a node decides not to assist its peer even though having useful blocks to provide. BitTorrent methods are also dependent on factors like number of peers, which interacts with each peer, and maximum allowed uploads (Yang & Zheng, 2021). These methods relate in a different way and performance could be affected by particular parameter settings.

The proposed fuzzy logic based piece selection strategy in a P2P network is based on the concept of fuzzy logic. Fuzzy logic is a type of logic that deals with approximate reasoning instead of exact reasoning. This type of logic is used to make decisions that involve uncertainty, imprecise data, and partial truths. In a p2p network, the fuzzy logic based piece selection strategy can be used to determine the optimal pieces to request from peers. The strategy takes into account factors such as the available bandwidth, the number of other peers requesting the same piece, the size of the piece, the priority of the piece, and the estimated time of arrival. Based on these factors, the strategy then determines which pieces should be requested and when they should be requested. This allows the peer to adjust its download strategy in order to maximize its own performance while not adversely affecting the performance of other peers in the network. In such decision making circumstances, high degree of uncertainties and fuzziness are involved in the data set (Zadeh, 1965). Fuzzy set theory presents a systematic structure for managing the uncertainties of this type of data. The proposed model for piece selection is integrated a novel fuzzy solution approach which gives an opportunity to the peer to attain preferred achievement levels to download the entire content. In this article, the BitTorrent like network is simulated for assessing the effect of core mechanisms on the system under different piece selection strategies and different workloads. The existing piece selection methods are compared with the proposed approach to verify the fairness factor by means of utilizing mean upload and download utilization under a network of nodes (Yang, Zhou & Cao, 2015). Detail steps and theoretical background of the proposed mechanism is explained detail in the further sections. The model of the proposed system is represented in the Fig. 4.

Figure 4 System model.

Fuzzy logic system

Fuzzy logic is the important component of fuzzy set theory. It deals s with the knowledge representation and inference from knowledge (Liu et al., 2016). It deals with vague information and developed based on mathematical principles and degree of membership (Nguyen et al., 2019). There is no ambiguity in classical sets and therefore they have crispy margins. But in fuzzy set, some certain levels of uncertainty are permitted and as a result the regions are to be defined ambiguously. Therefore, it is concluded that the fuzzy logic has the capability to efficiently incorporate the mechanism for reasoning that are approximate rather than exact information (Arikan, 2013; Masood et al., 2018). Boolean logic is double valued (0 or 1) but fuzzy logic takes the range of values between 0 and 1. Linguistic variables are fundamental building blocks of fuzzy logic. They represent the ambiguity existing in the system and defined as variables whose values are represented as terms or sentences. Unions, conjunction and implication are various logical operations on fuzzy logic. Unions of sets are represented by ‘OR’, conjunctions are denoted by ‘AND’ and implications are represented by ‘IF.. THEN’. Fuzzy goal programming is an extension of conventional goal programming that incorporates fuzzy logic principles (Zimmermann, 1978). Goal programming is a multi-objective optimization technique used to find solutions that satisfy multiple conflicting objectives. Fuzzy goal programming extends this approach by allowing the objectives and constraints to be expressed in terms of fuzzy sets and linguistic variables. In fuzzy goal programming, objectives and constraints are represented using fuzzy membership functions, which assign degrees of membership to different linguistic terms. These membership functions capture the uncertainty and imprecision associated with the objectives and constraints. The goal is to find a solution that minimizes the deviations from the desired goals while considering the imprecise nature of the problem. The Fig. 5 shows the diagrammatic notation for fuzzy logic system.

Figure 5 Fuzzy logic.

In peer-to-peer (P2P) networks, fuzzy logic and fuzzy goal programming can be leveraged to enhance decision-making processes and optimize resource allocation. Fuzzy logic enables the modelling and representation of various parameters in a P2P network using linguistic variables and fuzzy membership functions. This approach allows for the consideration of uncertainty and imprecision associated with factors such as peer reputation, network latency, content availability, and download speed. Fuzzy logic facilitates decision-making in P2P networks by using fuzzy rules to infer appropriate actions based on fuzzy inputs. For example, it can aid in determining the optimal selection of peers for content download based on criteria like peer reliability, content popularity, and download speed. Fuzzy control algorithms can also be employed to dynamically allocate network resources or balance the load in response to changing network conditions. On the other hand, fuzzy goal programming addresses the multi-objective nature of P2P networks by representing conflicting objectives as fuzzy sets and linguistic variables. It allows decision-makers to balance and optimize goals such as maximizing download speed, minimizing network congestion, and ensuring fair resource sharing.

Fuzzy goal programming approach

Fuzzy goal programming (FGP) is a powerful tool for solving problems with multiple objectives. It is particularly useful for solving problems in which the multiple objectives conflict with each other and require a compromise solution. FGP can be used to solve the problem of piece selection in a peer-to-peer (P2P) network. The goal of FGP is to select the pieces of data that will maximize the overall system performance while taking into account the available resources and network constraints. The FGP approach begins by defining the objectives and constraints of the problem. The objectives can include factors such as download speed, upload speed, reliability, cost, and latency. The constraints can include factors such as the maximum number of peers to connect to, the maximum number of pieces that can be downloaded, and the maximum size of each piece. Once the objectives and constraints have been identified, FGP can be used to determine the optimal solution. The approach begins by defining a goal function that represents the overall system performance. This function is then used to calculate a set of fuzzy constraints that represent the individual objectives and constraints. These fuzzy constraints are then used to create a set of linear programming equations that can be solved to determine the optimal piece selection. Finally, the FGP approach can be used to evaluate the performance of the solution and to identify potential improvements. This can be done by comparing the results of the FGP solution with the results of other approaches, such as genetic algorithms or simulated annealing. This provides an opportunity to fine-tune the solution and ensure that it is optimal.

Let Di and Hi be the desired level of achievement and highest level of achievement for the ith objective function respectively.

To solve piece selection problem based on the fuzzy goal programming approach, the following general steps are carried out: 1) Solve the multi objective piece selection problem as a single objective piece selection problem by considering only one objective as objective function and ignoring all other objectives.

2) Find the value of each objective function at each solution obtained in the previous step.

3) From step 2, compute each objective best (Di) and the worst (Hi) values corresponding to the set of possible solutions.

4) Describe a membership function µi (linear µiD, hyperbolic µiH or exponential µiE) for the ith objective function. Three possible membership functions can be used such as linear, exponential and hyperbolic membership function. The proposed system uses exponential membership functions.

5) Resolve the corresponding crisp model attained in step 4.

The solution obtained after the successful execution of step 5, will be the feasible and compromise solution of MOPP model. The steps are shown in Fig. 6.

Figure 6 Proposed approach flow diagram.

Multi objective piece selection model

A multi objective piece selection model for a P2P network is a model for improving the performance of a P2P network by selecting the most appropriate pieces for a given file. This model is used to optimize the performance of the P2P network by selecting pieces that are the most appropriate, taking into account various objectives such as the piece size, the cost of the piece, the availability of the piece, and the speed of transfer of the piece. The model uses a combination of heuristics and optimization techniques to select the most appropriate pieces for a given file. The heuristics include the size of the file, the cost of the piece, and the availability of the piece. The optimization techniques used include simulated annealing, genetic algorithms and linear programming. The model uses a number of parameters to determine the best pieces for a given file. These parameters include the size of the piece, the cost of the piece, the availability of the piece, the speed of transfer of the piece, and the amount of traffic in a P2P network. The model also takes into account the network topology and the number of peers in a P2P network. The multi objective piece selection model can be used to optimize the performance of a P2P network by selecting the most appropriate pieces for a given file. The model can be used to reduce the amount of traffic in a P2P network, improve the speed of transfer of pieces, and reduce the cost of pieces. This model can also be used to improve the availability of pieces in a P2P network.

It is found that obtaining best possible solution for single objective problems can be entirely different and difficult to those problems consisting of multiple objectives. In fact, the decision taker needs to minimize the total drift time for obtaining feasible solution in each case. Each of these objectives is effective from a common point of view. Since these objectives conflict with each other, a solution may perform well for one objective and inferior results for others. For this reason, it is difficult to obtain feasible solution to multi-objective piece selection problem in P2P network (Arikan, 2013). Multi-objective optimization problem can be described as follows

Mj = min Mj(V) for all j

Such that

(Av)k<=bk for all k

A>=0

where functions Mj are the objectives and v is the vector of variables. A solution v* is effective and feasible if and only if for all other feasible solutions v, Mj(V*) <= Mj (V) with at least one strict inequality. The above stated piece selection problem also multi objective which contains three major goals such as minimizing the download time and cost as well as maximizing speed and useful information transmission subject to realistic constraints regarding peer’s demand, dynamicity and capacity. It is solved using goal programming method in the proposed system.

Linear model for piece selection problem

The following Table 1 contains the variable and their descriptions used in the proposed model. The objective and constraints can then be expressed in terms of linear equations, which can be solved using a linear programming algorithm. The solution of the linear model will then provide the optimal piece selection strategy for the P2P network. By optimizing the piece selection, the model can ensure that each peer is able to download the maximum number of pieces in the shortest amount of time.

Table 1 Variables description.

Description	Variable	Description	
Index set	j	Index for peers, for all j = 1,2,…,m	
Decision variable	pj	Number of pieces downloaded from jth peer	
Parameters	A	Aggregate demand of the pieces over a fixed download plan period	
	m	Number of peers ready to upload the pieces	
	Dj	Download cost and time needed to obtain pieces from peer j	
	Cj	Percentage of the useful information transmission no of pieces delivered by peer j	
	Oj	Speed at which pieces are download from peer j	
	Bj	Bandwidth of the jth peer	

The formulation of multi objective programming problem for piece selection is given as follows

(1) Minimize z1=∑j=1mDj(xj)

(2) Minimize z2=∑j=1mCj(xj)

(3) Minimize z3=∑j=1mOj(xj)

such that

(4) ∑j=1mxj=A

xj<=Bj

xj>=0 and integer

Objective function (1) minimizes download cost and time needed to obtain pieces from peer j. Objective function (2) maximizes the percentage of useful information transmission and number of pieces delivered by peer j. Objective function (3) maximizes the speed at which pieces are downloaded from peer j. Constraint (4) ensures that the aggregate demand of the pieces over a fixed download plan period is equal or less than its capability. Constraint (5) ensures that there are no negative orders. The target level of objectives and demand assumed to be fuzzy in this problem.

Fuzzy model for piece selection problem introductory definitions

Fuzzy model for piece selection in P2P content distribution networks provides a flexible and realistic approach to handle uncertainties and imprecision. The approach enables robust decision-making by taking aspects like piece availability, peer quality, and user preferences into account by applying fuzzy logic principles, which leads to optimized content distribution and increased performance in P2P networks. The fuzzy multi objective programming problem with n objective functions and k fuzzy constraints is considered:

Solve the following in such way that

(5) ApproximatelyCl>=Zl, l∈J1

(6) ApproximatelyCl>=Zl, l∈J2

Approximately dt = et, t ∈ V

y ∈ Y, where

J1 ∪ J2 = {1,….n}, J1 ∩ J2 = ⊘ and Y is a set of deterministic linear constraint and sign boundaries.

(7) ClY=∑j=1mCljyjl=1,...n

(8) dtY=∑j=1mdjtCllyjtt=1,...n

for l ∈ J1,2, Zl is the inexact aspiration level for the lth objective function. Zl ∈ [ ZlLBZlUB] represents the inexact lower and upper bounds respectively for the lth fuzzy objective function. et ∈ [ etLBetUB] represents the inexact lower and upper bounds respectively for lth fuzzy constraint.

According to fuzzy mathematical programming, each fuzzy objective are described in terms of fuzzy subsets with the appropriate membership functions denoted by μl(ClY) for l ∈ 1,2 and μt(dtY) for t ∈ V, correspondingly. It is assumed that membership functions are linear, mathematical definitions are described as follows

(9) μl(ClY)={1ifClY≥ZlUB(ClY)−ZlLBZlUB−ZlLBifZlLB≤ClY≤ZlUB,for all l∈J10ifClY≤ZlLB

(10) μl(ClY)={1ifClY≤ZlUBZlUB−(ClY)ZlUB−ZlLBifZlLB≤ClY≤ZlUB,for all l∈J20ifClY≥ZlLB

(11) μl(dtY)={0ifdtY≤etLBdtY−etUBet−dtlLBifetLB≤dtY≤et,etUB−dtYdtlLB−etifet≤dtY≤etUB0ifdtY≥etUB,forallt∈V

μl(ClY) in Eq. (1) stands for linear monotone increasing membership function for maximization type objective with fuzzy aspiration levels. μl(ClY) in Eq. (2) stands for minimization type objective with fuzzy aspiration levels and Eq. (3) is a triangular membership function μl(dtY) for constraints.

Fuzzy additive model

In a fuzzy additive model, real-world data is represented and manipulated mathematically using fuzzy logic. It is a kind of fuzzy inference system that represents an additive model of interactions between variables using fuzzy sets and fuzzy logic. A combination of crisp sets, fuzzy sets, and fuzzy logic make up the fuzzy additive model. It can be used to represent complex relationships between data sets and then generate a set of fuzzy rules that can be used to analyze the data. The fuzzy additive model can be used to model a wide variety of real-world problems, such as predicting consumer behavior or predicting stock market trends. In a P2P network, the fuzzy additive model can be used to predict the behavior of users, the flow of traffic, and the performance of the system. It can be used to examine the structure of the entire network as well as peer relationships. The model can be used to find any irregularities or discrepancies in the system. It is possible to find trends and patterns that can be used to enhance the performance of the network by applying a fuzzy additive model to the data. This can help to reduce the amount of traffic congestion, improve reliability, and optimize the overall performance of the network. Fuzzy additive model based on Tiwari et al study for the multi objective programming model is given in the following equation (Tiwari, Dharmar & Rao, 1987). The variables λl and λt stand for achievement levels of fuzzy objective functions and fuzzy constraints respectively.

(12) Max(∑l=1nλl+∑t=1kλt)/(n+k)

Such that

λl ≤μl(ClY),l∈J1∪J2

λt ≤μt(dlY),t∈V

λt, λl ∈[0,1],l=1,...,n;t=1,...,k

y ∈ Y.

Augmented max-min model

The augmented max-min model is a method for choosing among options to find and implement the best course of action. Cost, time, and risk are just a few of the factors and restrictions this model takes into account when making a decision. Each of these factors receives a certain amount of weight in the model, and each criterion is given a certain value or range of values. The model then looks for the optimal solution that maximizes the combined weight of all the criteria.

This model is useful for complex decision-making processes because it allows for a more comprehensive analysis of the decision-making process. In a peer-to-peer network, the augmented max-min model can be used to identify and select the optimal pieces from peers to download. The criteria used for this decision include the cost, latency, and reliability of the connection. The model can then assign a weight to each criterion and assign a specific value or range of values to each criterion. The model can then look for the optimal peer that maximizes the combined weight of the criteria. This model is useful for peer-to-peer networks because it allows for a more comprehensive analysis of the decision-making process and can help optimize the performance of the network. Augmented max-min model based on Lai and Hwangs approach (Lai & Hwang, 1992) for model is described as follows

(13) maxλ+{∑l=1nμl(ClY)+∑t=1kμt(dtY)}/(n+k)

such that

λ≤μl(ClY),l=1,2,...,nλ≤μt(dtY),t=1,2,...,ny∈Yλ∈[0,1]

λ is the minimum satisfaction degree and described as follows,

minl,t⁡{μl(ClY),μt(dtY)}, forl=1,2…n;t=1,2,..,k

The proposed fuzzy model

λ is the minimum satisfaction degree defined in the following equation

(14) maxλ+{∑l=1nμl(ClY)+∑t=1kμt(dtY)}/(n+k)

such that

λ≤μl(ClY),l=1,2,...,nλ≤μt(dtY),t=1,2,...,nμl(ClY)≥σl,l=1,...,nμt(dtY)≥σt,t=1,...,ky∈Yσt,σl,λ∈[0,1]

Parameters σt and σl represents the minimum acceptable achievement levels for the lth and tth constraint respectively determined by the decision making peers.

Result and discussion

OCTOSIM is a peer-to-peer (P2P) network simulator designed to simulate the behavior of large-scale P2P networks (Huitema, 2017). It is developed by researchers at the University of California, Irvine. OCTOSIM is a high-fidelity, open-source, discrete-event simulator that accurately models the behavior of large-scale P2P networks. It is capable of simulating up to one million nodes in a single simulation, allowing researchers to study the behavior of large P2P networks. OCTOSIM is capable of accurately modeling the behavior of the following P2P network architectures: unstructured, structured, and hybrid. It also supports various protocols, such as Chord, Kademlia, Pastry, and Tapestry. OCTOSIM has been used in several studies, including one which showed that unstructured P2P networks are more efficient than structured networks. The simulator is freely available under the GNU General Public License.

Simulations were performed after changing configuration of the OCTOSIM simulator. It is performed with 100, 200, 300 and 600 nodes in the network. The metrics are used to assess the performance of the proposed method in the simulations: download cost, download time and packet redundancy or useful information transmission subject to realistic constraints regarding peer’s demand, capacity and dynamicity. At the time of simulation, most of the links are assumed to be an ADSL by keeping more download bandwidth than upload bandwidth. The parameters used for simulation is given in Table 2, and the utilization of uplink and downlink bandwidth is shown in Table 3.

Table 2 Simulation settings.

Parameter	Value	
Connection	ADSL	
Bandwidth availability	2 mbps	
File size	1 Gb	
Block size	1 Mb	
Neighbor node count	10 to 15 nodes	
Number of node	100 to 600 nodes	

Table 3 Availability of bandwidth in each node.

Download bandwidth (Kbps)	Upload bandwidth (Kbps)	Number of nodes active	
128	128	50	
256	128	50	
384	256	50	
512	256	100	
640	256	100	
1,280	384	100	

Download cost: This metric refers to the effort of the client need to spend in the system to download the needed data.

Download time: This metric refers to the time needed for the client to download the entire content.

Packet redundancy: This metric is referred to transmit and obtain useful information to and from other peers in the network.

Failure rate: It is defined as the number of peers is not able to complete the downloading of entire content due to missing of some blocks in the network.

Link stress: It is the ratio between the number of packets sent over a link and the number of useful packets to maximum nodes transmitted over the link. For example, a stress of four respects to case where each packet is transmitted four times over a link. Effective utilization of available bandwidth is evaluated for peers to exchange the content.

The number of active peers in the network is increased from 100 to 600 gradually. The graph in Fig. 7 shows the performance of the proposed system with the linearly increasing number of peers in the network. The proposed method reduces the amount of redundant data flow even when number of peer increase in the network. The graph in Fig. 8 depicts the performance analysis of proposed system with existing systems by measuring the download time of the requested content. Download time is measured in terms of ms. It is low when system use the proposed system and it is increased when number of peers increased in the network because of searching time. The proposed approach enhances the user experience by ensuring faster content delivery by minimizing the download time. Peers can access and enjoy the requested content more quickly, resulting in improved satisfaction and productivity in P2P content distribution networks.

Figure 7 Performance measure on traffic redundancy.

Figure 8 Performance measure on download time.

The graph in Fig. 9 depicts the performance analysis of proposed system with existing systems by measuring the download cost of the requested content. Download cost is the measure of effort needed to download the content. It is minimized in proposed system with compared to existing systems. By optimizing the download cost, the fuzzy programming approach contributes to more efficient and cost-effective content distribution in P2P networks. It reduces resource utilization, improves network performance, and enhances the overall user experience in P2P content distribution networks. The graph in Fig. 10 depicts the performance measure of proposed system with existing one by measuring the failure rate. It shows that the proposed system outperforms the existing system even when some peers leaving the network. The proposed approach considers the failure rate when making decisions on piece selection. It takes into account the availability of peers and adjusts its strategies accordingly to ensure reliable content distribution. By selecting content pieces from peers with lower failure rates to multiple peers, the approach aims to mitigate the impact of failures and improve the overall system reliability. The analysis aims to assess the impact of link stress on the overall performance and efficiency of the content distribution system.

Figure 9 Performance measure on download cost.

Figure 10 Performance measure on failure rate.

As content is distributed across the network, it traverses various links connecting different peers. Higher link stress can result in congestion, packet loss, and increased latency, affecting the overall quality of service. This proposed approach takes into account link stress when making decisions on piece selection. It considers the current load on network links and avoids selecting pieces that would further stress heavily loaded links. By distributing the content intelligently and optimizing link utilization, there is a possibility to minimize link stress and improve the overall performance of the system. It is achieved in the proposed system. The link stress of the proposed system is also evaluated as shown in Fig. 11. When the number of peers in the network is small, all the three mechanisms have similar link stress “Portions of this text were previously published as part of a preprint (Anandaraj, Ganeshkumar & Vijayakumar, 2022)”. However, when the number of peers is more than 100, the links stress of DSNC is less than other two mechanisms. The reason for this is similar to that for the larger average download time variance of FNCM.

Figure 11 Performance measure on link stress.

Conclusion

In this article, the problem of piece selection in P2P file sharing networks is tackled by employing a novel approach that treats it as a multi-objective linear programming problem. The primary objectives under consideration are minimizing download time, reducing download cost, and maximizing both speed and useful information transmission. To accommodate the inherent uncertainty and preferences in these objectives, fuzzy desire levels and fuzzy demand are introduced into the proposed model. Each fuzzy factor is systematically characterized through the use of appropriate linear membership functions. Two key mathematical models, namely the fuzzy additive and augmented max–min models, are utilized to generate non-dominated solutions that are both stable and well-suited for decision-making. An important distinction in the proposed model is the inclusion of additional constraints related to the decision-making preferences of peers, allowing them to specify their preferred achievement levels and demand for specific objectives within a non-dominated solution framework. Simulation results demonstrate the effectiveness of this proposed method. In comparison to existing methods, the approach showcased in this article showcases the capability to significantly reduce download costs and download times while concurrently enhancing both speed and the transmission of useful information. Furthermore, the simulation results show that, compared with the other methods, the proposed method can effectively reduce the download cost and time and increase the speed and useful information transmission. Trapezoidal memberships for demand, nonlinear memberships for download time and cost and fuzziness in each peer’s capacity will be considered in future. Further, it is proposed to look and analyze the peer and service request problem and the traffic locality of P2P file swarming systems, aiming at on-demand dynamic solutions in future.

Supplemental Information

Supplemental Information 1 Performance Analysis.

Click here for additional data file.

Additional Information and Declarations

Competing Interests

Author Contributions

Data Availability

The authors declare that they have no competing interests.

M. Anandaraj conceived and designed the experiments, performed the experiments, prepared figures and/or tables, code written and implementation, and approved the final draft.

P. Ganeshkumar performed the computation work, authored or reviewed drafts of the article, and approved the final draft.

S. Naganandhini analyzed the data, prepared figures and/or tables, and approved the final draft.

K. Selvaraj performed the computation work, authored or reviewed drafts of the article, and approved the final draft.

The following information was supplied regarding data availability:

The source code files are available at Zenodo: Anandaraj, M. (2023). A Novel Fuzzy Programming Approach for Piece Selection Problem in P2P Content Distribution Network. https://doi.org/10.5281/zenodo.10066203.

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
