# Peer review of "A novel fuzzy programming approach for piece selection problem in P2P content distribution network"

_PeerJ Computer Science, doi:10.7717/peerj-cs.1645_

## Round 0.1 · original submission · Major Revisions

Please follow review comments, perform requested corrections, and resubmit a revised version

Reviewer 1 ·

Basic reporting

• Manuscript is clear, relevant for the field and presented in a well-structured manner.
• Cited references are not mostly recent publications (within the last 5 years) and relevant, does not include an excessive number of self-citations (I will add some details in the end).
• Manuscript sound scientifically.
• Figures and schemes are appropriate, show the real data, are easy to interpret and understand.
• Conclusions are consistent with the evidence and arguments presented.
• Dear authors, please update your references where possible, they should not be older then 5-7 years.

Experimental design

no comment

Validity of the findings

The data on which the conclusions are available in an acceptable discipline-specific repository. The data in my opinion is robust, statistically sound, and controlled.

Additional comments

no comments

Reviewer 2 ·

Basic reporting

3. Improve the structure of the introduction.
4. Following research articles can be added in your manuscript:
o Khabbaz, Maurice, Chadi Assi, and Sanaa Sharafeddine. "Multihop V2U path availability analysis in UAV-assisted vehicular networks." IEEE Internet of Things Journal 8.13 (2021): 10745-10754.
o Islambouli, Rania, et al. "Towards trust-aware IoT hashing offloading in mobile edge computing." 2020 International Wireless Communications and Mobile Computing (IWCMC). IEEE, 2020.
o Rahman, Sawsan Abdul, et al. "Internet of things intrusion detection: Centralized, on-device, or federated learning?." IEEE Network 34.6 (2020): 310-317.
o Hassan, Muhammad Abul, et al. "Intelligent Transportation Systems in Smart City: A Systematic Survey." 2023 International Conference on Robotics and Automation in Industry (ICRAI). IEEE, 2023.

Experimental design

1. Explicitly describe the pros and cons of each existing method

Validity of the findings

2. The paper is readable but needs improvement in terms of grammar, suitable words, sentence selection, etc.

Reviewer 3 ·

Basic reporting

No Comments

Experimental design

No Comments

Validity of the findings

No comments

Additional comments

Comments:
I appreciate the opportunity to contribute to the evaluation of this important document. It is an excellent opportunity to learn and share academic knowledge.
The authors proposed a fuzzy programming approach that this article to solve the multiple objectives piece selection problem in P2P network. The topic of this study is significant for the Content Delivery Networks. The motivation behind the study was presented clearly. The surveys on CDN and P2P are sufficient. However, the paper is not ready for publication due to the following reasons:
1. The paper is poorly organized, and it is not clear what exact problem the paper is addressing.
2. The contribution of this study is unclear. It is difficult to understand the limitations of existing works and the difference between the proposed work and other existing works.
3. The validation of the proposed approach is insufficient.

Here are the detailed comments:
1.
The motivations behind the specific design of the proposed approach are not clear.
2.
The authors should consider using health-related data for their study. The MNIST dataset used in the study is not relevant to e-healthcare systems.
3.
Author should polish the titles at all levels of the paper, highlight the hierarchy of the paper, and correspond to the technical route of the introduction.
4. Improve grammatical and typos-related issues of the manuscript.
5. Authors should outline the significant contribution to the knowledge at the end of section 1 or 2
6. The literatures required improvement by identify problem, method, solution and results in logical presentation
7. In the end of Introduction Section, the Section distribution should be modified along with contributions.
8. The organization of this paper is good but needs much work on few points, such as the general discussion and general information must be avoided, especially in the introduction section. The authors need to focus on the main challenges and the main problem of the research.
9. Improve Figures for example Fig 1, and some other text (caption) is displayed mismatch.
10. Remove general formulas and add your proposed approach based mathematical description.
11. In the references try to include latest (2020 - 2022) references and style and arrange in ascending order.
12. Review your result section, need more accurate explanation. Also, check mathematical expression.
13. Explain more the close and open interval

---

## Round 0.2 · Minor Revisions

Follow review comments and submit a revised version

Reviewer 1 ·

Basic reporting

No comment.

Experimental design

No comment, everything is really good.

Validity of the findings

No comment.

Additional comments

This work was really good before (to my opinion) but now the authors did the great work adding all experimental results files, updated references list and added adequate description so I do not have any additional improvements to add.

Reviewer 2 ·

Basic reporting

Authors have incorporated all previous reviewers comments and this paper can be accepted in current form.

Experimental design

Good

Validity of the findings

Novel findings

Additional comments

N/A

Reviewer 3 ·

Basic reporting

All points of major revision have been addressed.

Experimental design

It is updated now.

Validity of the findings

Validated.

Additional comments

1) In the references try to include latest (2022-2023) references.
2) Add system model diagram along with proposed workflow.
3) Precise the conclusion.
4) Improve the English Language.

---

## Round 0.3 · accepted · Accept

The authors have addressed all of the reviewers' comments.